# Invasive Species Change Plant Community Composition of Preserved Prairie Pothole Wetlands

**DOI:** 10.3390/plants12061281

**Published:** 2023-03-11

**Authors:** Seth A. Jones, Edward S. DeKeyser, Cami Dixon, Breanna Kobiela

**Affiliations:** 1School of Natural Resource Sciences, North Dakota State University, Fargo, ND 58105, USA; 2Chase Lake National Wildlife Refuge, U.S. Fish and Wildlife Service, Woodworth, ND 58496, USA

**Keywords:** plant invasions, *Bromus inermis* Leyss., *Phalaris arundinacea* L., *Typha ×glauca* Godr. (pro sp.) [angustifolia or domingensis × latifolia], invasion dynamics, responses of native species

## Abstract

Plant communities in North American prairie pothole wetlands vary depending on hydrology, salinity, and anthropogenic disturbance in and around the wetland. We assessed prairie pothole conditions on United States Fish and Wildlife Service fee-title lands in North Dakota and South Dakota to improve our understanding of current conditions and plant community composition. Species-level data were collected at 200 randomly chosen temporary and seasonal wetland sites located on native prairie remnants (*n* = 48) and previously cultivated lands that were reseeded into perennial grassland (*n* = 152). The majority of species surveyed appeared infrequently and were low in relative cover. The four most frequently observed species were introduced invasive species common to the Prairie Pothole Region of North America. Our results suggested relative cover of a few invasive species (i.e., *Bromus inermis* Leyss., *Phalaris arundinacea* L., and *Typha ×glauca* Godr. (pro sp.) [angustifolia or domingensis × latifolia]) affect patterns of plant community composition. Wetlands in native and reseeded grasslands possessed distinct plant community composition related to invasive species’ relative cover. Invasive species continue to be prevalent throughout the region and pose a major threat to biological diversity, even in protected native prairie remnants. Despite efforts to convert past agricultural land into biologically diverse, productive ecosystems, invasive species continue to dominate these landscapes and are becoming prominent in prairie potholes located in native areas.

## 1. Introduction

The Prairie Pothole Region (PPR) includes >770,000 km^2^ through the north–central United States and portions of central Canada [1,2]. This formerly glaciated area is famous for millions of “prairie pothole” wetlands [3]. The PPR experiences variable temperatures, precipitation, evapotranspiration rates, topography, and land-uses across latitudinal and longitudinal gradients [1,4,5]. A substantial portion of prairie potholes has been lost to anthropogenic disturbance (i.e., conversion to agricultural land use), but >2.6 million remain, comprising roughly 26,000 km^2^ [6].

Prairie potholes possess unique hydrologic, biotic, chemical, and physical characteristics and are typically classified based on water permanency and vegetation zonation [7,8,9]. Pothole types often differ not only in vegetation zonation and water permanence but also in abundance, area, landscape position, cover type, and physiochemical properties. Prairie potholes harbor many native plant species, provide wildlife habitat, and support other ecosystem services, including flood mitigation, filtration of pollutants, groundwater recharge, nutrient retention, water for livestock, and recreational opportunities [10,11,12,13,14]. Throughout the PPR, wetland plant community composition can be used to determine wetland condition (i.e., ecological integrity), which, in turn, points to a pothole’s ability to provide important ecosystem services [15,16].

A wide variety of disturbances occur throughout the PPR, ranging from historically/naturally occurring disturbances (e.g., fire, grazing, and climatic variability) to anthropogenic disturbances (e.g., cultivation, drainage, urbanization, ditching, sedimentation, and chemical runoff [17,18,19]). The conversion of native prairie to croplands has been one of the most significant driving factors degrading potholes during the past 150 years [17,20], with up to 90% of temporary and seasonal wetlands lost in certain areas [21]. Smaller wetlands experience shorter periods of inundation and can, therefore, be manipulated relatively easily and converted to crop production [22] when water is not present on the surface. Thus, many of the smaller prairie pothole wetlands were converted to croplands throughout the PPR, especially those that were relatively accessible or located within flatter landscapes. Alterations to hydrology (drainage, ditching, and runoff diversion) are among the most severe direct anthropogenic disturbances negatively affecting potholes and their plant communities [18,20]. In addition, changes in potholes’ hydrology related to climate change have also been shown to affect vegetation composition [23,24]. Specifically, climate change and land use disturbances have been shown to increase the abundance of invasive species and facilitate their spread [18,22,24,25]. Increased abundance of invasive species consequently reduces biodiversity and ecological productivity [26,27,28,29,30,31]. In addition, many formerly cultivated public lands, now planted into perennial grasslands, were reseeded with invasive species, which has also facilitated the spread and dominance of invasive species [32,33].

Wetland management by the United States Fish and Wildlife Service (USFWS) in the PPR began with the acquisition of land after the 1934 passing of the Duck Stamp Act [32], a piece of legislation that ultimately led to the establishment of National Wildlife Refuges and the acquisition of other fee-title lands (e.g., Waterfowl Production Areas). The current vegetation composition of many of these fee-title lands is influenced by earlier management strategies that relied on the widespread seeding of certain introduced species, which remain dominant to this day [34]. The seeding of introduced species, coupled with the idle grassland management strategies employed at the time, was partially responsible for the spread of non-native species to other areas throughout the PPR [32]. By the 1990s, management strategies shifted toward the inclusion of fire, grazing, and diverse native mixes for reseeding in an effort to preserve and improve the region’s diverse native plant communities that evolved under background disturbance regimes that included fire and grazing. However, most of the USFWS fee-title land reseedings focused on upland plant species and relied on the natural colonization of wetland plant species in the wetland areas.

Pothole plant communities differ greatly between wetlands residing in natural or remnant native grasslands and those in reseeded grasslands [22,35,36,37]. Galatowitsch and van der Valk [22] found that most native species are unlikely to reestablish themselves in restored wetlands or are outcompeted by invasive species after re-establishment. Similarly, Seabloom and van der Valk [37] found differences in plant community composition and the dominance of invasive species between restored and natural wetlands. Native perennials were associated almost exclusively with natural wetlands, while introduced and disturbance-tolerant species were associated with restored wetlands. While there are many differences in plant community composition between natural and restored wetlands, the greater relative abundance of invasive species in restored wetlands is particularly concerning.

Invasive species have a variety of mechanisms that allow them to influence plant community composition, including native patch suppression [38], seedling resource suppression [25,39,40], and soil nutrient and microbe modification [41]. Invasive species often have a greater tolerance to non-natural disturbance than native species, therefore becoming even more dominant in highly disturbed environments [42,43,44,45]. Whether invasive species are established through intentional reseeding or natural processes, there are mechanisms by which they can affect their associated plant communities. Invading species have been shown to affect resident plant communities to a similar extent, whether they were introduced through intentional seeding or by chance through natural dispersal [46].

Three of the most problematic invasive plant species in the PPR are *Bromus inermis* Leyss. [28], *Phalaris arundinacea* L. [47], and *Typha ×glauca* Godr. (pro sp.) [angustifolia or domingensis × latifolia] [33]. (Taxonomic nomenclature follows the United States Department of Agriculture PLANTS database [48]). *Bromus inermis*, *P. arundinacea,* and *T. ×glauca* are each most commonly associated with the low prairie, wet meadow, and shallow marsh wetland vegetation zones, respectively [7]. While other cattail species are still believed to be prevalent in the PPR, genetic analyses have shown *T. ×glauca* to be the most abundant [49]. Despite the low prairie zone not truly being considered a wetland plant community, the low prairie zone is often included in assessments and analyses due to its proximity to the wet meadow zone, the often-observed overlap in plant species, and because the low prairie experiences less plant community variation due to local weather conditions [7,15,16,50].

Management objectives throughout the PPR have evolved over several decades to emphasize diversity and ecological integrity [32] by incorporating prescribed burns and livestock grazing in an attempt to reproduce historical disturbance regimes. While knowledge of how to best manage wetlands in the PPR continues to improve, more information on pothole plant communities is necessary to make proper management decisions. In this study, we survey wetland plant communities of 200 prairie potholes located on USFWS fee-title land in the southern PPR (in North Dakota and South Dakota, USA) to provide insight into the current wetland plant community composition and determine potential factors influencing plant community composition. Previous research has led us to believe that wetland plant community composition in native grasslands would be different from that in reseeded grasslands, with invasive species being especially common and comprising a large portion of relative cover in reseeded grasslands. Plant communities were expected to be influenced by a combination of species-specific drivers (e.g., relative cover of *T. ×glauca*), environmental factors (e.g., salinity), and site history (native grassland versus reseeded grassland).

## 2. Results

### 2.1. Species Frequency

In total, 348 plant species representing 207 genera were identified in 200 wetlands, of which 60 were annuals, 13 were biennials, 275 were perennials, 285 were native, and 63 were introduced. Not all introduced species are considered invasive—this varies depending on the region the species is in and the definition of invasive used. Most species observed over the course of this study were found infrequently. For example, 301 species were found in fewer than 25% of the wetlands assessed. In contrast, 25 species were present in at least 50% of wetlands (Table 1), and 47 species were encountered in 25% of sites. Two species, *Cirsium arvense* (L.) Scop. and *Poa pratensis* L., both introduced, were present at all 200 sites. The four most frequently occurring species (*P. pratensis*, *C. arvense*, *Bromus inermis*, and *Sonchus arvensis* L.) were invasive introduced species commonly found in the low prairie and wet meadow zones of PPR wetlands ([7,50] Table 1). These four species were found in >90% of the wetlands assessed.

Native species accounted for 16 of the 25 most frequently occurring species, but the majority of these were species known to tolerate disturbance ([50,51], Table 1). None of the 25 most frequently occurring species was considered a species intolerant of disturbance. However, 104 of the 348 species observed over the course of the study were species considered disturbance-intolerant.

### 2.2. Wetlands Located in Native versus Reseeded Grassland

Multi-response permutation procedure (MRPP) indicated that plant community compositions of the wet meadow zones of temporary wetlands located on native grasslands were distinct from those located on reseeded grasslands (*p* = 0.003). In contrast, MRPP did not indicate distinct plant community composition for low prairie zones (*p* = 0.208) of temporary wetlands located on native and reseeded grasslands. For seasonal wetlands, MRPP indicated plant community composition was different for all vegetation zones on native and reseeded grasslands (i.e., low prairie (*p* < 0.001); wet meadow (*p* < 0.001); shallow marsh (*p* < 0.001)). Differences in plant community composition for the plant communities likely stem from the differences in the abundance of invasive species between wetlands in native and reseeded grassland.

### 2.3. Temporary Wetlands

After the low prairie and wet meadow relative cover datasets for temporary wetlands were trimmed to exclude species that appeared in fewer than 5% (i.e., three wetlands) of the 59 temporary wetlands included in this study, 63 species remained in the low prairie dataset and 43 species remained in the wet meadow dataset. Non-metric multi-dimensional scaling (NMS) analysis of the temporary wetlands low prairie dataset produced a three-dimensional solution (stress = 14.08, 62 iterations, instability = 0.00047; Figure 1a). The three axes of the NMS ordination of the low prairie dataset cumulatively accounted for 84.6% of the variation in the relative cover dataset (Axis 1 = 41.1%, Axis 2 = 22.5%, and Axis 3 = 20.9%). Four species were correlated (Pearson r ≥ |0.4|) with Axis 1 of the low prairie NMS, five species with Axis 2, and eight species with Axis 3 (Table 2).

*Bromus inermis* relative abundance was determined to be the most influential driver determining site position in species space along Axis 1 of the low prairie ordination as *B. inermis* possessed the strongest correlation (r = −0.91) and was the only species significantly negatively correlated with Axis 1 (Table 2). *Bromus inermis* was present as a primary species in 54 of 59 temporary wetlands’ low prairie zones, often comprising 50–100% of relative cover, especially in reseeded grasslands (Figure 1a). The species correlated with Axes 2 and 3 of the NMS ordination (Table 2) did not appear to indicate clear drivers of the temporary wetland low prairie plant communities due to the weaker strength of the correlations and less variance explained by NMS Axes 2 and 3.

Non-metric multi-dimensional scaling analysis of the temporary wetlands wet meadow dataset produced a three-dimensional solution (stress = 16.84, 65 iterations, instability = 0.00048; Figure 1b). The three axes of the NMS ordination of the wet meadow dataset cumulatively accounted for 73.1% of the variation in the wet meadow plant communities (Axis 1 = 29.2%, Axis 2 = 26.2%, and Axis 3 = 17.7%). Two species were significantly correlated with Axis 1 of the wet meadow NMS, four species with Axis 2, and two species with Axis 3 (Table 2). *Phalaris arundinacea* relative abundance was determined to be the most influential driver for NMS Axis 1 in temporary wetlands’ wet meadow zones as *P. arundinacea* was strongly negatively correlated (r = −0.92) with the axis. *Phalaris arundinacea* was present as a primary species in 32 of 59 wet meadows of temporary wetlands, with relative cover often approaching 100% where present (Figure 1b). The species correlations with Axes 2 and 3 of the NMS ordination (Table 2) did not appear to indicate clear drivers of the temporary wetlands’ wet meadow plant communities due to the weaker strength of the correlations and less variance explained by NMS Axes 2 and 3.

### 2.4. Seasonal Wetlands

The low prairie, wet meadow, and shallow marsh relative cover datasets for seasonal wetlands were trimmed to exclude species that appeared in fewer than 5% (i.e., seven wetlands) of the 141 seasonal wetlands included in this study. Excluding rare species resulted in 63 species in the low prairie dataset, 56 species in the wet meadow dataset, and 28 species in the shallow marsh dataset.

Non-metric multi-dimensional scaling analysis of the seasonal wetlands low prairie dataset produced a three-dimensional solution (stress = 17.1565, 77 iterations, instability = 0.0004; Figure 2a). The three axes of the NMS ordination of the low prairie dataset cumulatively accounted for 80.1% of the variation (Axis 1 = 39.1%, Axis 2 = 22.5%, and Axis 3 = 18.5%). Two species were correlated (|r| ≥ 0.40) with Axis 1 of the low prairie NMS, nine species with Axis 2, and two species with Axis 3 (Table 3). *Bromus inermis* was determined to be the most influential driver of Axis 1 of the low prairie NMS as it was strongly negatively correlated (r = −0.90) with the axis and the only species significantly negatively correlated with this axis (Table 3). *Bromus inermis* was present as a primary species in 133 of 141 low prairies of seasonal wetlands, often comprising 50–100% relative cover, especially in reseeded grasslands (Figure 2a). The species correlations with Axes 2 and 3 of the NMS ordination did not appear to indicate clear drivers of the seasonal wetlands’ low prairie plant communities due to the weaker strength of the correlations and less variance explained by NMS Axes 2 and 3 (Table 3).

Non-metric multi-dimensional scaling analysis of the seasonal wetlands wet meadow dataset produced a three-dimensional solution (stress = 17.58, 73 iterations, instability = 0.00048; Figure 2b). The three axes of the NMS ordination of the wet meadow dataset cumulatively accounted for 75.7% of the variation in the wet meadow plant communities (Axis 1 = 41.7%, Axis 2 = 20.3%, and Axis 3 = 13.7%). Two species were correlated with Axis 1 of the wet meadow NMS, two species with Axis 2, and two species with Axis 3 (Table 3). *Phalaris arundinacea* was determined to be the most influential driver for Axis 1 of the wet meadow NMS as it was strongly negatively correlated (r = −0.93) with the axis. *Phalaris arundinacea* was present as a primary species in 94 of 141 wet meadows of seasonal wetlands, with relative cover often approaching 100% when present, especially in reseeded grasslands (Figure 2b). Species correlations with Axes 2 and 3 of the NMS ordination did not appear to indicate clear drivers of the seasonal wetlands’ wet meadow plant communities due to the weaker strength of the correlations and less variance explained by NMS Axes 2 and 3 (Table 3).

Non-metric multi-dimensional scaling analysis of the seasonal wetland shallow marsh dataset produced a three-dimensional solution (stress = 13.70, 95 iterations, instability = 0.0005; Figure 2c). The three axes of the NMS ordination of the shallow marsh dataset cumulatively accounted for 82.9% of the variation in the shallow marsh communities (Axis 1 = 42.7%, Axis 2 = 25.2%, and Axis 3 = 15%). Four species were correlated with Axis 1 of the shallow marsh NMS, two species with Axis 2, and two species with Axis 3 (Table 3). *Typha ×glauca* was determined to be the most influential driver for Axis 1 of the shallow marsh NMS as it possessed the strongest correlation (r = −0.88) and was the only species significantly negatively correlated with the axis. *Typha ×glauca* was present as a primary species in 93 of 141 shallow marshes of seasonal wetlands, comprising nearly 100% relative cover where present, especially in reseeded grasslands (Figure 2c). The species correlations with Axis 2 show species potentially following a pattern of hydrologic regime, but Axes 2 and 3 of the NMS ordination do not appear to indicate clear drivers of the seasonal wetlands’ shallow marsh plant communities (due to the weaker strength of the correlations and less variance explained by NMS Axes 2 and 3; Table 3).

## 3. Discussion

The results of this study provide an examination of the effects invasive species have on prairie pothole wetland plant communities and can aid in future research and/or management intended to improve the diversity and ecological integrity of USFWS fee-title lands. The emphasis was on detecting the most frequently occurring species on USFWS fee-title lands, exploring differences in plant communities of wetlands located within native and reseeded grasslands, and determining the major drivers influencing plant community composition.

Four species (*P. pratensis*, *C. arvense*, *B. inermis*, and *S. arvensis*) were detected in >90% of the temporary and seasonal wetlands. The four most frequently occurring species were invasive introduced species commonly found in the low prairie and wet meadow zones of prairie potholes [7,50]. This indicates that these four invasive species are now present within wetlands located in native prairie areas and are no longer limited to wetlands in highly disturbed or reseeded areas.

Conversely, the majority of species observed in this study occurred infrequently; 301 of 348 species were found in <25% of wetlands surveyed. This is likely in part a reflection of the large study area but also due to limited wetland surveys conducted in native grassland areas, where there was much higher diversity than for wetlands in reseeded grasslands. These results correspond with previous research showing that a high percentage of total species diversity can be found at very few sites (e.g., wetlands located in native grassland or forested areas of the PPR) and that certain plant communities and species within wetlands are highly sensitive to disturbance and often underrepresented in restored wetlands [18,35,52,53]. Few species intolerant of disturbance [51] were widespread throughout the study and typically comprised a minor component of relative cover where present, largely due to the influence invasive species have on the plant communities [31,52,53,54].

The low prairie zone of temporary wetlands was the only plant community surveyed where there was no difference in plant community composition between wetlands located on native and reseeded grasslands. Multi-response permutation procedure indicated differences between wetlands located in native and reseeded grassland for all zones of seasonal wetlands and the wet meadow zone of temporary wetlands. Low prairie zones of temporary wetlands located on native and reseeded grasslands are likely equally impacted by invasive species throughout the region surveyed in the current study (e.g., *B. inermis* and *P. pratensis* relative cover was similar for wetlands on native and reseeded grasslands).

The results of this study corroborate previous studies showing that plant community composition differs between native and reseeded wetlands [36,37,53,55]. These differences are commonly attributed to former anthropogenic and agricultural disturbances in reseeded grasslands. Past disturbances (e.g., cultivation) can give invasive species a competitive advantage [18,56] and often allows them to form monocultures [33]. Due to the large geographical area included in this study, it cannot be assumed that the 152 reseeded sites were subjected to identical anthropogenic and agricultural disturbances (nor should it be assumed that all anthropogenic disturbances were absent from the 48 native sites). However, despite the variation in land use history and extent of anthropogenic and agricultural disturbance, invasion by non-native species was widespread, indicating the remarkable ability of invasive species to influence plant communities, regardless of disturbance history, once these species have a presence on the landscape.

*Bromus inermis* relative cover appeared to be the most significant driver of low prairie plant community composition. *Bromus inermis* has previously been shown to influence plant community composition in the PPR. Grant et al. [31] showed that a high frequency and cover of *B. inermis* was most prevalent in areas of low native species cover and vice versa in the same study area. This also confirms the results of other previous studies comparing native versus invasive frequency and cover [27,28]. *Bromus inermis* is known to dominate and spread and, therefore, to restrict and suppress the growth of other species [38]. This invasion allows it to completely take over entire areas of grassland and edge out any remaining pockets of higher diversity. Using these mechanisms of invasion, *B. inermis* appeared to be outcompeting all other species in the low prairie zone, invasive species included. Previous research has shown that dominant invasive species can influence the abundance of native species and other invasive species alike [26], similar to how species were suppressed by *B. inermis* in this study. *Poa pratensis* has long been viewed as a major threat to plant community biodiversity [27,28,31], but it is evident that *B. inermis* is currently the greatest threat in the low prairie zone of the native and reseeded temporary and seasonal wetlands included in this study. *Poa pratensis*, while present at all 200 wetlands sites, did not seem to reduce floristic diversity and was often suppressed by *B. inermis*.

*Phalaris arundinacea* was identified as the major driver of wet meadow communities. Similar to *B. inermis* in the low prairie, the relative cover of *P. arundinacea* explained more variation in the wet meadow datasets than any other factor. *Phalaris arundinacea* has many different mechanisms of invasion, allowing it to control and negatively impact plant community composition [57,58]. *Phalaris arundinacea* exhibits rapid growth, self-facilitation, an ability to handle disturbance, and the suppression of native seedlings, allowing the species to become dominant [47,59]. Mullhouse and Galatowitsch [52] showed invasive perennials, particularly *P. arundinacea*, frequently occurred in restored (i.e., previously cultivated) wetlands, with cover approaching 75–100%, resulting in the absence of many common native wetland species. These results are similar to our findings, with *P. arundinacea* often nearing 100% relative cover in wet meadows on reseeded grassland (i.e., previously cultivated). Reseeded areas are often still surrounded by agricultural land, where the surface water runoff may be accompanied by higher nitrogen levels, which tend to increase the abundance of *P. arundinacea* and further facilitate its competitive advantage in these areas [43,47].

Similar to all other plant communities tested, the major driver for the shallow marsh zone was the invasive species *T. ×glauca*. *Typha ×glauca* has been known to form monocultures in wetlands [33], significantly lower wetland diversity [25,55], and is often most prevalent in disturbed wetlands [56]. Other variables, such as weather patterns, may have affected the plant community composition of the shallow marsh [23]. The timing of the site visit could affect the plant communities being observed as seasonal wetlands tend to reach a drawdown phase by the end of the growing season. Year 2 of the study was also significantly drier than Year 1, which could have led to a greater abundance of drawdown species and a lower abundance of species that require hydrology [7,23]. However, none of these factors would change the influence *T. ×glauca* has on the shallow marsh plant communities. Drawdown phase versus emergent open water phase trends were seen in species correlation coefficient patterns along Axis 2 of the NMS ordination [7], but Axis 2 explained less variation in the dataset than Axis 1, which was driven by *T. ×glauca* abundance. *Phalaris arundinacea* often reached high levels of relative cover in the shallow marsh when *T. ×glauca* was sparse or absent or in areas where it was unlikely to grow. This is likely due to *P. arundinacea*’s ability to withstand varying levels of hydrology and soil saturation [60]. Still, *P. arundinacea* did not have nearly as high of a correlation in the shallow marsh as *T. ×glauca* and did not exhibit the same level of influence on plant community composition as it does in the wet meadow zone, reinforcing the observation that the most dominant invasive species is the most influential driver of plant community composition in each wetland zone. This again reinforces that the dominant invasive species displaces both native species and other less dominant invasive species alike [26].

Previous research has shown the influence dominant invasive species can have on plant community diversity, species abundance, and overall composition [26,27,28,29,30,31,38]. This study has determined the main drivers of overall plant community composition to be the abundance of a particular invasive species within each vegetation zone (i.e., *B. inermis* in the low prairies, *P. arundinacea* in the wet meadows, and *T. ×glauca* in the shallow marshes), regardless of hydrology (i.e., temporary or seasonal wetlands).

It is commonly found that invasive plant species can drive plant community composition subcategories such as plant species diversity and abundance [26,29,30]. Non-metric multi-dimensional scaling ordination results are often used to show patterns or relationships among plant communities but can be used to determine drivers of plant community composition [61,62]. Rahman et al. [62] determined various environmental variables as composition drivers, while Kahmen et al. [61] showed species with specific traits such high productivity, competitive ability, and nutrient use efficiency to be potentially significant drivers of community composition and productivity.

*Bromus inermis* was commonly used in the reseeding process prior to updated management regimes [32], in part helping to explain why the species is widespread throughout the wetlands surveyed despite their location on native or reseeded grasslands. While widespread planting of *B. inermis* has certainly facilitated its spread, planting alone cannot account for the species’ role as a driver of plant community composition. *Bromus inermis* was not always used in the reseeding process and would have to outcompete other species in the seed bank or those trying to reestablish themselves through natural dispersal at all sites, including wetlands located on native grasslands. Petermann et al. [46] showed that invasive species can affect plant communities to a similar extent, whether they were intentionally seeded (e.g., *B. inermis*) or established themselves through natural dispersal (e.g., *P. arundiacea* and *T. ×glauca*), and our NMS results can help corroborate that.

Currently, invasive species appear to have taken over much of the previously cultivated FWS fee-title land, regardless of the mechanism of introduction to the landscape, and are encroaching into many areas of native grassland. While native prairie wetlands remain relatively intact with high native species diversity and cover, invasive species encroaching into these wetlands have the potential to significantly influence native plant communities. Maintaining plant species diversity and the ecological integrity of native prairie wetlands should take the highest priority as these areas harbor a high percentage of native species despite their relatively small area. Prevention of increased abundance seems a more realistic goal than completely transforming wetland plant communities where invasive species already dominate.

Future research is necessary to better understand the influence invasive species have on prairie pothole plant communities. Continual monitoring of sites will be necessary to obtain information regarding plant community changes over time, especially in native prairie wetlands. Further analysis could be conducted to aid in understanding wetland plant community composition and drivers. Multivariate analyses should be performed with native prairie wetlands and reseeded prairie wetlands separated to explore how patterns of community composition differ between the two. Additional analyses could be conducted to corroborate our interpretation of invasive species as drivers of the plant communities. Further examination of the datasets could also show which native species can coexist most easily with a high cover of invasive species. Knowing this will be useful for maintaining some level of diversity at sites where invasive species cannot be controlled.

In order to maintain biodiversity and ecological integrity throughout the PPR, it is crucial to not only focus future research on the best ways to manage invasive species but also to recognize which other introduced species are likely to cause similar issues in the future. Qi et al. [30] showed the influence invasive species have on plant community diversity, but the decline in diversity was not observed until invasive species cover was beyond 10%, indicating the decrease in diversity may not be apparent immediately and some invasive species may be overlooked because they are disregarded early in the invasion process. Our data shows certain introduced species that are often not thought of as highly invasive, such as *Phleum pratense* L. or *Alopecurus arundinaceus* Poir., are exhibiting invasive tendencies and have very high cover at some sites. Some of these species are used regionally in seeding mixes and to provide forage for wildlife and livestock, potentially providing an avenue for these species to spread throughout the region. Van Kleunen et al. [63] provide a model showing the naturalization of introduced species in relation to their economic use. Plants introduced for animal feed have high rates of naturalization success. This model, along with cover data gathered, could be used to help predict which species are becoming highly naturalized and may cause major problems for species diversity and ecosystem integrity in the future.

## 4. Materials and Methods

### 4.1. Site Selection and Study Area

Wetland sample sites were selected using spatial data layers imported into a GIS environment and delineated across state and USFWS fee-title land boundaries [64]. A generalized random tessellation stratified sampling design was used to generate a randomly selected but spatially balanced distribution of wetland sites stratified by hydrologic regime (as determined by the National Wetland Inventory (NWI)) and sample year using the “spsurvey” package in R [65,66,67,68]. Wetlands located entirely within the boundaries of USFWS fee-title land in North Dakota, South Dakota, or Montana were deemed available for this study. Site selection was further constrained to only include temporarily and seasonally ponded wetlands, resulting in 125 temporary and 125 seasonal wetland sites, to undergo further evaluation for potential inclusion as field sites. The first 100 randomly generated sites were designated as primary sample sites, with an additional 25 “oversample” sites reserved as potential sites to be substituted if the primary sites were deemed not appropriate for sampling by further evaluation. The results of the random site selection reflected the relative abundance of potholes throughout the region, with 157 in North Dakota, 91 in South Dakota, and 2 in Montana. The selection also reflected the regional land use history of USFWS fee-title lands, with 176 located within reseeded grasslands (i.e., areas that had been cultivated at some point in the past) and 74 located within native remnant prairie (i.e., areas that were not cultivated).

After the random selection of 250 potential wetland sites, each was visually inspected using aerial imagery by a team of experts from the USFWS, U.S. Geological Survey, and North Dakota State University [64] prior to a field visit. Visual inspections were conducted in sequential order, beginning with the first randomly generated wetland site (see [63] for a detailed description of site selection and determination). The final classification as a temporary or seasonal wetland was determined by field crews during a site visit and based on vegetation zonation apparent at the wetland site. Temporary wetlands possessed both a low prairie and wet meadow vegetation zone. Seasonal wetlands possessed low prairie, wet meadow, and shallow marsh vegetation zones. All wetlands classified as temporary or seasonal by field crews were retained for the study, even if the field determination differed from the original NWI classification. Field crews would substitute an oversample site when field visits indicated that a randomly selected site was not a wetland (i.e., actually upland), not accessible (e.g., surrounded by private land), or the wetland possessed more permanent hydrology than seasonal (e.g., semi-permanent or permanent wetlands), which ultimately resulted in the inclusion of 59 temporary wetlands and 141 seasonal wetlands located in North Dakota and South Dakota (Figure 3).

### 4.2. Field Sampling

There were 152 wetlands surveyed in reseeded grasslands and 48 in native grasslands. Wetland site visits were conducted in June–August of 2020 and 2021. Weather conditions differed between the two years. Temperature was above average across the study area for both years of the study but was higher in 2021 than in 2020 [69]. Precipitation was below average for both years, but drought conditions were more severe across the study area in 2021. Palmer Drought Severity Index (PDSI) values were observed as low as −6.6 during the field season in the study area in 2021 and never reached below −1.0 for 2020.

All plants were identified to the species level whenever possible by field crews conducting surveys following a quadrat method [15,16]. Two plant identification field guides were primarily used for the field identification of plant species [70,71]. When species could not be identified in the field, samples were collected to be examined and identified in the North Dakota State University Herbarium. Aerial cover by species was estimated using 1 m^2^ quadrats dispersed throughout each vegetation zone (Figure 4) and relativized. Vegetation zones were analyzed separately due to the associated differences in physical parameters and species composition [7,15,16]. Scientific names, common names, origin (Native or Introduced), and life-history guild (Annual, Biennial, Perennial) were taken from the U.S. Department of Agriculture’s PLANTS Database [48].

### 4.3. Data Analysis

Wetland plant community compositions were organized and explored in relation to plant species’ characteristics (e.g., physiognomy, native or introduced origin, etc.) and frequency of occurrence at wetland sites. Species that occurred in 50% or more of the sites were recorded to give a better understanding of species frequency for the fee-title lands of North and South Dakota. Multivariate analysis was conducted using PC-ORD version 7 software [72,73]. Prior to multivariate analysis, species datasets were transformed using the arcsine square root transformation and trimmed to ensure data were not skewed by an overabundance of zeros by excluding species that occurred in <5% of wetland sites [73,74]. Excluding rare species from the multivariate analyses helps to focus the multivariate analysis on overall plant community composition rather than the presence or absence of relatively rare species [74] that may be encountered throughout the large geographic area included in this study. Multi-response permutation procedure (MRPP) was used to compare the plant community composition of wetlands located in native and reseeded grasslands for each vegetation zone using the relative Sørenson distance measure, following Smith et al. [54].

Non-metric multi-dimensional scaling (NMS) was conducted using species relative cover estimates at each site to examine seasonal and temporary wetland plant communities, following Smith et al. [54]. Species’ correlations (Pearson r) with the NMS axes were considered to indicate a species’ role as a potential driver of the various wetland plant communities. Species with Pearson correlation coefficients of r ≥ |0.4| were selected for further examination as significant drivers. Absolute r values were interpreted as an indication of a species’ relative strength as a driver of wetland plant community composition along an NMS axis. Thus, species with the largest absolute r values were determined to be the drivers of the corresponding NMS axes for the various wetland plant communities. In instances where several species possessed similar r values, species correlations were considered to represent a trend among species rather than to depict a clear driver of the NMS axis. In addition, because a decreasing amount of information inherent in the dataset is explained by each successive NMS axis, emphasis was placed on the first axis of each NMS solution.

## 5. Conclusions

This study provides the baseline assessment necessary to guide future research and management and contributes to the understanding of wetland plant community composition and drivers of that composition for fee-title lands in the region. Land managers must understand plant community composition to achieve management goals related to diversity and ecological integrity. Prairie pothole wetlands are dynamic ecosystems undergoing constant change due to natural and anthropogenic disturbances, ultimately affecting plant community composition [18,23].

The results indicate invasive species are influencing the plant community composition of PPR wetlands as they have been shown to previously influence uplands and, therefore, should thus be the focus of future management [28,31,45]. The most frequently detected species, as a whole and in each separate plant community, were invasive species. Whether wetland plant communities located on native or reseeded grasslands differed largely depended on the abundance of invasive species. Analyzing the species datasets using NMS ordination confirmed the role specific invasive species are playing in influencing wetland plant community composition. In all of the plant community datasets evaluated, the axis explaining the most variation in the dataset was driven by the most dominant invasive species in the plant community (*B. inermis* for the low prairie, *P. arundinacea* for the wet meadow, and *T. ×glauca* for the shallow marsh). Taking all of these factors into account, it could be inferred that invasive species as a whole are playing a critical role in shaping the wetland plant communities of both native and reseeded grasslands in the PPR. The factors affecting their abundance and mechanisms in which they drive the plant communities may vary, but they appear to be the major drivers nonetheless.

## Figures and Tables

**Figure 1 plants-12-01281-f001:**
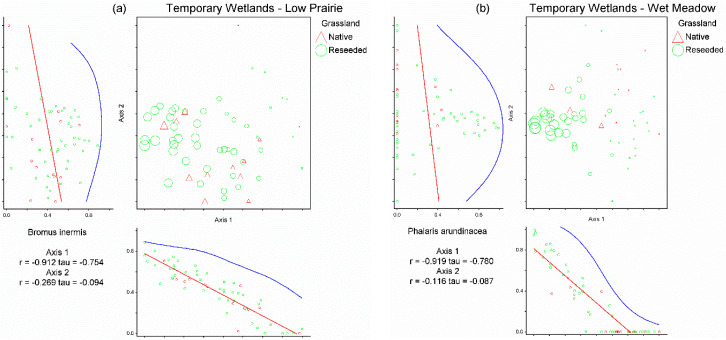
Non-metric multi-dimensional scaling ordinations for the low prairie vegetation zone (**a**) and wet meadow vegetation zone (**b**) of 59 temporary wetlands. Each triangle represents a temporary wetland located on native grassland and each circle represents a temporary wetland located on reseeded grassland. The relative size of site symbols (triangles/circles) indicates the relative cover of *Bromus inermis* in the low prairie zone (**a**) and *Phalaris arundinacea* in the wet meadow zone (**b**). Species’ correlations with each axis are included beneath the horizontal axis and to the left of the vertical axis (i.e., *B. inermis* in the low prairie vegetation zone (**a**); *P. arundinacea* in the wet meadow vegetation zone (**b**)).

**Figure 2 plants-12-01281-f002:**
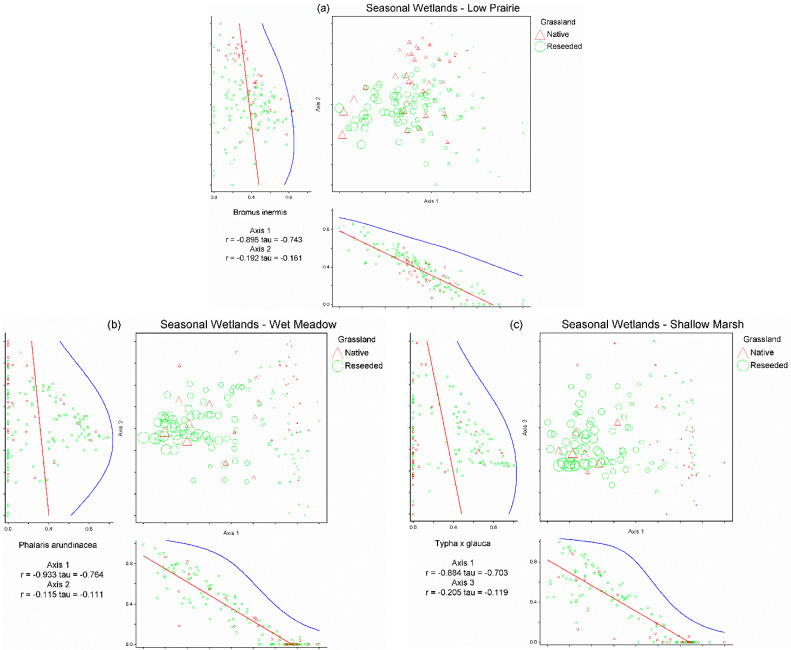
Non-metric multi-dimensional scaling ordinations for the low prairie vegetation zone (**a**), wet meadow vegetation zone (**b**), and shallow marsh vegetation zone (**c**) of 141 seasonal wetlands. Each triangle represents a seasonal wetland located on native grassland and each circle represents a seasonal wetland located on reseeded grassland. The relative size of site symbols (triangles/circles) indicates the relative cover of *Bromus inermis* in the low prairie zone (**a**), *Phalaris arundinacea* in the wet meadow zone (**b**), and *Typha ×glauca* in the shallow marsh zone (**c**). Species’ correlations with each axis are included beneath the horizontal axis and to the left of the vertical axis (i.e., *B. inermis* in the low prairie vegetation zone (**a**), *P. arundinacea* in the wet meadow vegetation zone (**b**), *T. ×glauca* in the shallow marsh zone (**c**)).

**Figure 3 plants-12-01281-f003:**
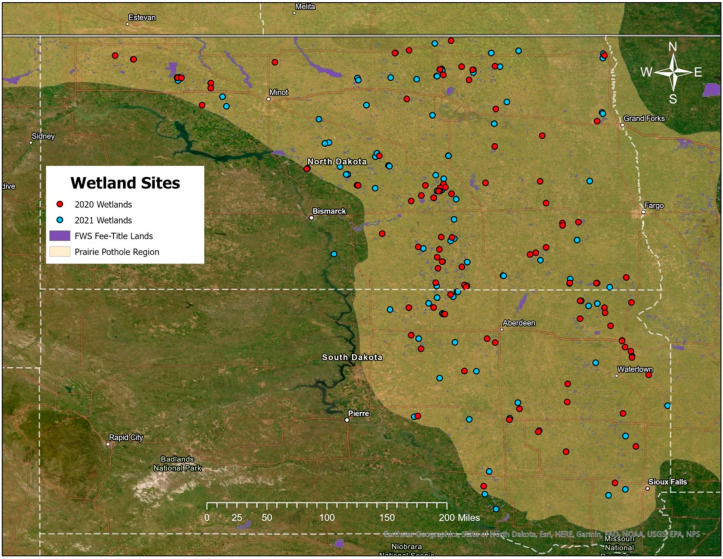
Distribution of the 200 temporary and seasonal wetlands sampled, with the southern Prairie Pothole Region and North Dakota and South Dakota outlined. Sites visited in 2020 are shown in red, and sites visited in 2021 are shown in blue.

**Figure 4 plants-12-01281-f004:**
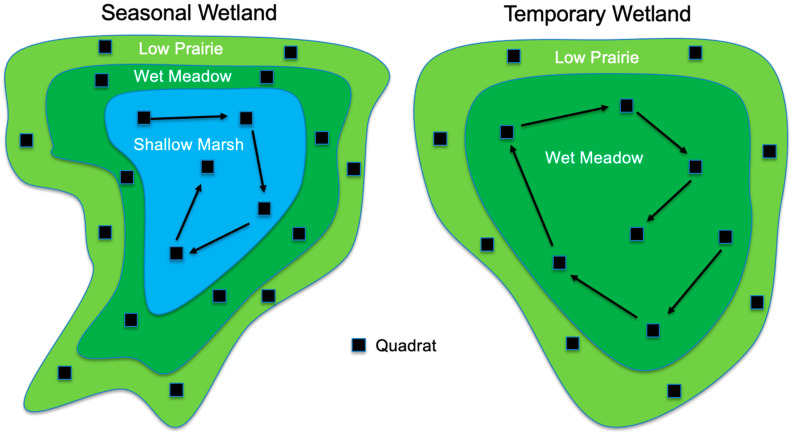
Example quadrat layouts by vegetation zone for seasonal and temporary wetlands following DeKeyser et al. [15] and Hargiss et al. [16]. Quadrats (1 m^2^) in the exterior zones (low prairie and/or wet meadow zones) were placed near the center of the zone, with random but relatively even spacing around the wetland. Quadrats within the interior zone (shallow marsh for seasonal wetlands and wet meadow for temporary wetlands) were placed starting at the outer edge of the zone and progressing in a spiral fashion toward the center of the wetland.

**Table 1 plants-12-01281-t001:** Species identified in at least 50% of the wetlands surveyed.

Species	Origin	N ^a^
*Cirsium arvense*	Introduced	200
*Poa pratensis*	Introduced	200
*Bromus inermis*	Introduced	195
*Sonchus arvensis*	Introduced	184
*Symphyotrichum lanceolatum*	Native	174
*Solidago canadensis*	Native	167
*Rumex crispus*	Introduced	150
*Polygonum amphibium var. stipulaceum*	Native	150
*Carex pellita*	Native	149
*Asclepias speciosa*	Native	149
*Phalaris arundinacea*	Native *	145
*Elymus repens*	Introduced	145
*Symphyotrichum ericoides*	Native	130
*Symphoricarpos occidentalis*	Native	123
*Spartina pectinata*	Native	122
*Rosa woodsii*	Native	117
*Glycyrrhiza lepidota*	Native	114
*Carex atherodes*	Native	111
*Typha ×glauca*	Introduced	110
*Eleocharis palustris*	Native	110
*Stachys pilosa*	Native	109
*Melilotus officinalis*	Introduced	109
*Hordeum jubatum*	Native	108
*Anemone canadensis*	Native	107
*Artemisia absinthium*	Introduced	103

^a^ Number of wetlands in which the species was present. * Native species but considered invasive in Prairie Pothole wetlands, and there are introduced varieties that have hybridized with native species.

**Table 2 plants-12-01281-t002:** Plant species evaluated as potential drivers of plant community composition of the low prairie and wet meadow vegetation zones of temporary wetlands. Species with Pearson correlation coefficients of r ≥ |0.4| with an NMS axis were selected for further examination as significant drivers. Absolute r values were interpreted to indicate a species’ relative strength as a driver of wetland plant community composition. Emphasis was placed on the first NMS axis because a decreasing amount of variation within the community dataset is explained by each additional axis.

Species	Low Prairie			Wet Meadow		
	Axis 1 ^a^	Axis 2 ^a^	Axis 3 ^a^	Axis 1 ^a^	Axis 2 ^a^	Axis 3 ^a^
*Achillea millefolium*			−0.410			
*Anemone canadensis*		−0.477				
*Artemisia ludoviciana*		−0.504	0.434			
*Bromus inermis*	−0.912					
*Calamagrostis canadensis*						−0.521
*Carex laeviconica*						−0.702
*Elaeagnus commutata*			0.417			
*Eleocharis palustris*					0.516	
*Elymus repens*	0.516			0.521		
*Galium boreale*			0.422			
*Hordeum jubatum*					0.635	
*Pascopyrum smithii*		0.441				
*Phalaris arundinacea*				−0.919		
*Poa pratensis*		0.610	0.558			
*Polygonum amphibium var. stipulaceum*					−0.611	
*Ratibida columnifera*			−0.474			
*Rosa woodsii*			0.402			
*Rumex crispus*					0.600	
*Solidago canadensis*	0.402					
*Symphoricarpos occidentalis*		−0.509	0.529			
*Symphyotrichum lanceolatum*	0.446					

^a^ Pearson correlation with NMS axes.

**Table 3 plants-12-01281-t003:** Plant species evaluated as potential drivers of plant community composition of the low prairie, wet meadow, and shallow marsh vegetation zones of seasonal wetlands. Species with Pearson correlation coefficients of r ≥ |0.4| with an NMS axis were selected for further examination as significant drivers. Absolute r values were interpreted to indicate a species’ relative strength as a driver of wetland plant community composition. Emphasis was placed on the first NMS axis because a decreasing amount of variation within the community dataset is explained by each additional axis.

Species	Low Prairie			Wet Meadow			Shallow Marsh		
	Axis 1 ^a^	Axis 2 ^a^	Axis 3 ^a^	Axis 1 ^a^	Axis 2 ^a^	Axis 3 ^a^	Axis 1 ^a^	Axis 2 ^a^	Axis 3 ^a^
*Anemone canadensis*		0.494							
*Artemisia ludoviciana*		0.424							
*Bromus inermis*	−0.895								
*Calamagrostis canadensis*					0.537				
*Carex atherodes*							0.511	0.617	
*Carex pellita*						0.709			
*Elaeagnus commutata*		0.463							
*Eleocharis palustris*									0.417
*Elymus repens*	0.506								
*Galium boreale*		0.487							
*Helianthus pauciflorus*		0.430							
*Lemna turionifera*								−0.526	
*Melilotus officinalis*		−0.409							
*Phalaris arundinacea*				−0.933			0.412		0.545
*Poa pratensis*		0.594	−0.567						
*Polygonum amphibium var. stipulaceum*						−0.569	0.594		
*Rosa arkansana*		0.457							
*Solidago canadensis*			0.428						
*Spartina pectinata*				0.509	−0.651				
*Symphoricarpos occidentalis*		0.560							
*Typha ×glauca*							−0.884		

^a^ Pearson correlation with NMS axes.

## Data Availability

Raw data and datasets used for analyses are available upon request from the corresponding author.

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
