# Peer review of "Invasive Species Change Plant Community Composition of Preserved Prairie Pothole Wetlands"

_plants, 2023, doi:10.3390/plants12061281_

Round 1

Reviewer 1 Report

The paper is well written and has interesting results, it deserves publication in the Plants journal after having made the following minor changes:

- correctly follow the rules of the taxonomic nomenclature code, at least

the first time a species is mentioned in the text it must be written with the complete scientific binomial.

- Authors should indicate which work is referred to for the nomenclature of taxa so as not to create confusion while reading (eg, Bromus inermis Leyss. is considered the basinym of Bromopsis inermis (Leyss.) Holub

Furthermore, for Phalaris arundinacea the subspecies referred to should be indicated.

- the figures in the text are not high resolution, they need to be improved

All the best.

Author Response

The paper is well written and has interesting results, it deserves publication in the Plants journal after having made the following minor changes:

- correctly follow the rules of the taxonomic nomenclature code, at least the first time a species is mentioned in the text it must be written with the complete scientific binomial.

  • This suggestion has been incorporated throughout the revised manuscript.

- Authors should indicate which work is referred to for the nomenclature of taxa so as not to create confusion while reading (eg, Bromus inermis Leyss. is considered the basinym of Bromopsis inermis (Leyss.) Holub

  • Taxonomic nomenclature follows the United States Department of Agriculture PLANTS database, now indicated in the text of the revised manuscript (line 98).

- Furthermore, for Phalaris arundinacea the subspecies referred to should be indicated.

  • No subordinate taxa are recognized by the United States Department of Agriculture PLANTS database

- the figures in the text are not high resolution, they need to be improved

  • The figures have been modified to conform to journal requirements.

Reviewer 2 Report

What is the relative cover of the invasive species Cirsium arvense, Poa pratensis, Sonchus arvensis? The authors state only that it is low. But which one exactly?

The first time you mention it, you should point out that Typha ×glauca is the result of hybridization (=T. angustifolia x T. latifolia).

Author Response

Comments and Suggestions for Authors

What is the relative cover of the invasive species Cirsium arvense, Poa pratensis, Sonchus arvensis? The authors state only that it is low. But which one exactly?

  • The language was clarified throughout the manuscript to better convey that the discussion of these species was related to presence/absence instead of cover.

The first time you mention it, you should point out that Typha ×glauca is the result of hybridization (=T. angustifolia x T. latifolia).

  • Revised to “Typha ×glauca Godr. (pro sp.) [angustifolia or domingensis × latifolia]” (as recognized by the United States Department of Agriculture PLANTS database) at first mention.

Reviewer 3 Report

 Are the regional disturbances to Prairie potholes (lines 45-52.) affecting some/all or those in the study or do those disturbances apply to destroyed Prairie Potholes?  In general, what other land use regimes were associated with the Potholes prior to the study sites before these became fee-title land?  What management practices, if any, are associated with the current ownership/land use? Are all these services management priorities for the current owners (Lines 39-41)?

Line 70-74. It is unclear how this impacts the study sites.  Also, there appears to be either missing or an extra word in line 73.

Lines 105-106. What exactly did these management changes entail?  Better explanation of these circumstances will help make this work useful to scientists from other parts of the world.

Is there any factor that determined whether a pothole was disturbed and reseeded versus left undisturbed?  Was conversion sufficiently “random” or are there any underlying tendencies to consider?

Lines 290-292.  What types of disturbances are meant here?  You reference cultivation earlier in the paragraph. What does “not all communities where disturbed to the same extent” mean?   More years of row cropping or different techniques?  Cultivation versus some other unnamed disturbance?  Was there a particular type of crop/crops associated with “cultivation”?  This is important given particular herbicide and fertilizer inputs or the timing of soil disturbance activities.

Line 366 delete “it”

Line 403-405. If a species is commonly seeded, is that the means of entry into the site or is it naturalized and spreading, suggesting invasiveness?

Line 405. Does “forage” imply a grazing land use in these landscapes?  If so, is pasture improvement considered a component of cultivation or something else?

Author Response

Comments and Suggestions for Authors

 Are the regional disturbances to Prairie potholes (lines 45-52.) affecting some/all or those in the study or do those disturbances apply to destroyed Prairie Potholes?  In general, what other land use regimes were associated with the Potholes prior to the study sites before these became fee-title land?  What management practices, if any, are associated with the current ownership/land use? Are all these services management priorities for the current owners (Lines 39-41)?

  • Language has been added to the introduction, methods, and discussion to provide a bit more detail about past disturbances and current management throughout the region and specific to the study sites.

Line 70-74. It is unclear how this impacts the study sites.  Also, there appears to be either missing or an extra word in line 73.

  • These statements were revised to better convey the impact to the study area. The extra word was removed.

Lines 105-106. What exactly did these management changes entail?  Better explanation of these circumstances will help make this work useful to scientists from other parts of the world.

  • More detail was added to better illustrate the relevance of the management changes.

Is there any factor that determined whether a pothole was disturbed and reseeded versus left undisturbed?  Was conversion sufficiently “random” or are there any underlying tendencies to consider?

  • Details have been added to the introduction section to address these points.

Lines 290-292.  What types of disturbances are meant here?  You reference cultivation earlier in the paragraph. What does “not all communities where disturbed to the same extent” mean?   More years of row cropping or different techniques?  Cultivation versus some other unnamed disturbance?  Was there a particular type of crop/crops associated with “cultivation”?  This is important given particular herbicide and fertilizer inputs or the timing of soil disturbance activities.

  • This paragraph was revised to better make the connections between species invasion and land use history/management.

Line 366 delete “it”

  • This sentence was reworded in the revised text.

Line 403-405. If a species is commonly seeded, is that the means of entry into the site or is it naturalized and spreading, suggesting invasiveness?

  • These statements were revised and clarified to make the link from seeding to spread/potential naturalization and invasiveness.

Line 405. Does “forage” imply a grazing land use in these landscapes?  If so, is pasture improvement considered a component of cultivation or something else?

  • These statements were revised to better describe an example of regional land use.

Round 2

Reviewer 3 Report

No additional comments